# Solvation of Large Polycyclic Aromatic Hydrocarbons in Helium: Cationic and Anionic Hexabenzocoronene

**DOI:** 10.3390/molecules27196764

**Published:** 2022-10-10

**Authors:** Miriam Kappe, Florent Calvo, Johannes Schöntag, Holger F. Bettinger, Serge Krasnokutski, Martin Kuhn, Elisabeth Gruber, Fabio Zappa, Paul Scheier, Olof Echt

**Affiliations:** 1Institut für Ionenphysik und Angewandte Physik, Universität Innsbruck, 6020 Innsbruck, Austria; 2Laboratoire Interdisciplinaire de Physique, CNRS, Université Grenoble Alpes, F-38000 Grenoble, France; 3Institut für Organische Chemie, Universität Tübingen, 72076 Tübingen, Germany; 4Laboratory Astrophysics and Cluster Physics Group of the MPI for Astronomy, University of Jena, Helmholtzweg 3, 07743 Jena, Germany; 5Department of Physics, University of New Hampshire, Durham, NH 03824, USA

**Keywords:** helium, PAH, hexabenzocoronene, adsorption, mass spectrometry, path-integral molecular dynamics simulations

## Abstract

The adsorption of helium on charged hexabenzocoronene (Hbc, C_42_H_18_), a planar polycyclic aromatic hydrocarbon (PAH) molecule of *D*_6*h*_ symmetry, was investigated by a combination of high-resolution mass spectrometry and classical and quantum computational methods. The ion abundance of He*_n_*Hbc^+^ complexes versus size *n* features prominent local anomalies at *n* = 14, 38, 68, 82, and a weak one at 26, indicating that for these “magic” sizes, the helium evaporation energies are relatively large. Surprisingly, the mass spectra of anionic He*_n_*Hbc^−^ complexes feature a different set of anomalies, namely at *n* = 14, 26, 60, and 62, suggesting that the preferred arrangement of the adsorbate atoms depends on the charge of the substrate. The results of our quantum calculations show that the adsorbate layer grows by successive filling of concentric rings that surround the central benzene ring, which is occupied by one helium atom each on either side of the substrate. The helium atoms are fairly localized in filled rings and they approximately preserve the *D*_6*h*_ symmetry of the substrate, but helium atoms in partially filled rings are rather delocalized. The first three rings contain six atoms each; they account for magic numbers at *n* = 14, 26, and 38. The size of the first ring shrinks as atoms are filled into the second ring, and the position of atoms in the second ring changes from hollow sites to bridge sites as atoms are filled into the third ring. Beyond *n* = 38, however, the arrangement of helium atoms in the first three rings remains essentially frozen. Presumably, another ring is filled at *n* = 68 for cations and *n* = 62 for anions. The calculated structures and energies do not account for the difference between charge states, although they agree with the measurements for the cations and show that the first solvation shell of Hbc^±^ is complete at *n* = 68. Beyond that size, the adsorbate layer becomes three-dimensional, and the circular arrangement of helium changes to hexagonal.

## 1. Introduction

In vacuum, helium nanodroplets (HNDs) rapidly cool to approximately 0.37 K by evaporative cooling. At this temperature, they are superfluid [1,2]. However, when an atomic or molecular ion is embedded in a HND or bulk helium, the density of the helium in its immediate surrounding will be enhanced due to electrostriction, and a solid-like sphere of helium, dubbed a snowball, is likely to surround the ion [3,4].

The basic features of Atkins’ snowball model, which is based on a classical continuum approach, have been confirmed by several authors [5,6]. However, the microscopic features of the snowball depend on several factors, last but not least, on the nature of the solvated ion. Obviously, its size will affect the number of atoms in the first solvation shell. Furthermore, the ion may favor or disfavor the formation of an ordered helium layer. Ar^+^, for example, will be surrounded by three well-ordered solvation shells of icosahedral symmetry containing 12, 20, and 12 atoms, respectively [7,8]. For Pb^+^, on the other hand, the approximately 15 helium atoms in the first solvation shell are highly disordered, and there is no clear distinction between atoms in the first and second solvation shell [9].

Even starker differences appear when molecular ions are embedded in helium. The corrugated surface of C_60_ cations or anions favors the formation of an ordered, commensurate layer containing 32 helium atoms, each atom residing in the hollow of a pentagonal or hexagonal carbon ring [10,11,12,13]. The first solvation shell, however, is not complete until C_60_^±^ ions are surrounded by 60 He atoms [14]. Another aspect appears if one studies size-selected He*_n_*C_60_^+^ ions: In He_32_C_60_^+^, the helium atoms are strongly localized, but they become delocalized when atoms are removed from or added to the commensurate layer [12,15,16].

The convex, strongly corrugated surface of fullerenes favors the localization of adsorbed helium atoms; linear molecules such as the acenes and planar two-dimensional polycyclic aromatic hydrocarbons (PAHs) offer weaker corrugation. Acenes essentially confine a single helium atom to one dimension. A single adsorbed atom is spatially delocalized along the long axis of the molecule, with negligible hindrance by the corrugation of the aromatic microsurface [17,18]. In contrast, three helium atoms adsorbed on one side of tetracene are predicted to form a quasi-one-dimensional trio that occupies a quantum coherent but highly correlated ground state [19].

The planar coronene molecule (Cor, C_24_H_12_, *D*_6*h*_ point group) offers a quasi two-dimensional surface for the adsorption of helium. A basic sphere packing model would predict that the growth of the adsorbate layer occurs via successive filling of ring-shaped subshells that preserve the symmetry of the molecule. Assuming that, for even-numbered sizes *n*, an equal number of helium atoms will reside above and below the molecular plane, one might expect that the first two He atoms will reside in the hollow above and below the central benzene ring (we will call this a (1 + 1) configuration). Adding 6 + 6 atoms in the hollows of the outer benzene rings would fill the first ring at *n* = 14. The next ring would be filled at *n* = 26 upon adding 6 + 6 atoms in the open dimples between the outer H-free C atoms and C–H bridges, and another ring would be filled at *n* = 38 upon adding another 6 + 6 atoms at a slightly larger distance from the molecular *C*_6_ axis in the dimples between the external C–C bonds [20].

A classical study of neutral He*_n_*Cor does, in fact, show an arrangement of the adsorbate atoms as expected, and enhanced stability for *n* = 14, 26, 38 (for brevity, we will refer to sizes that enjoy enhanced stability as “magic numbers”) [21]. However, the enhanced stability of clusters with *n* = 14 and 26 disappears when quantum effects are taken into account. The mass spectra of cationic He*_n_*Cor^+^ indicate enhanced stability for *n* = 38 but not for 14 and 26; additional magic numbers were seen for *n* = 41 and 44 [20]. A classical study of He*_n_*Cor^+^ ions predicted a magic number at *n* = 38 while a quantum study by means of path-integral molecular dynamics (PIMD) simulations predicted slightly enhanced stability for *n* = 38, and strongly enhanced stability for *n* = 41 and 44. At larger sizes, the dynamics of the helium atoms become increasingly fluxional [22].

The failure of expectations based on geometric packing is partly due to the energetic cost (i.e., the zero point energy) that comes with the confinement of helium atoms, and partly due to the mismatch between the length parameter of the corrugated surface (which has adjacent hollow sites spaced 0.246 nm apart) and the size of the helium atom (approximately 0.3 nm). These effects push the helium atoms toward the periphery, and the helium atoms in the first ring do not reside at the centers of the benzene rings.

Another planar PAH that we have recently studied by mass spectrometry and PIMD simulations is triphenylene (Tpl, C_18_H_12_, *D*_3*h*_ point group). After filling the hollow site above and below the central ring, one would naively expect a magic number at *n* = 8 with the addition of 3 + 3 atoms. Instead, a magic number was found at *n* = 6 in experiments and theory because the quantum effects and the size mismatch between the template and the helium atom favor a vacancy at the central benzene ring [23,24,25]. Furthermore, the central ring is not aromatic and has low electron density, in contrast to the three outer rings [26,27]. A species seeking electron density should preferably bind to the three outer rings instead of the central one.

Here, we investigate the adsorption of helium on cationic and anionic hexabenzocoronene (Hbc, C_42_H_18_, *D*_6*h*_ point group). Hbc is a PAH consisting of thirteen fused benzene rings. It has numerous potential applications including organic light emitting diodes, solar cells, field effect transistors, and, if synthesized into a porous organic polymer, as an adsorbent for volatile organic compounds [28,29,30]. Furthermore, PAHs as large as Hbc strongly absorb the vacuum ultraviolet photons emitted by hot stars; they are believed to be stable in the interstellar medium with respect to dehydrogenation and to be ionized, even reaching the dicationic state [31,32,33].

The large size of Hbc prompts us to address two questions: (i) How does the relaxed confinement in sparsely covered He*_n_*Hbc^±^ change the arrangement of the helium atoms and the stability (or magic numbers) of the first few subshells? and (ii) How many helium atoms fit into subshells beyond *n* = 38? Furthermore, this is the first study of a negatively charged PAH complexed with helium. Rather unexpectedly, we observed that the magic numbers depend on the charge state of the ion. The only magic numbers common to both charge states were *n* = 14 and a weaker anomaly at *n* = 26. Further prominent magic numbers in the mass spectra of cations were *n* = 38, 68, 82, and, for anions, *n* = 60, 62, and around 84.

To assist with the interpretation of these measurements, we carried out atomistic modeling of the He*_n_*Hbc^±^ cations and anions using the same computational methodology as in our earlier studies [10,23,24,34,35]. The Hbc^±^ ion is treated as rigid, with its geometry optimized at the level of density-functional theory (DFT). Optimization by basin-hopping was performed to identify low-energy structural candidates for 1 to 100 He atoms attached to Hbc^±^. Nuclear delocalization was then included through the PIMD methodology. The simulations essentially predict the same magic numbers for cations and anions, namely *n* = 14, 24, 38, and at 68 upon closure of the first solvation shell. The agreement with the experimental values is thus fair for the cations, but the calculations cannot account for the differences observed with the anions. However, they do support the notion that the adsorbate layer grows by successively filling rings containing 6 + 6 atoms each. With few exceptions, the He atoms in filled rings are localized, but rather delocalized in partially filled rings. Moreover, while the radii of the rings initially contract as the adsorbate layer grows, adding He beyond *n* = 38 keeps the geometry of the three innermost rings essentially unchanged.

## 2. Materials and Methods

### 2.1. Experimental Methods

Hexa-peri-hexabenzocoronene (Hbc, C_42_H_18_, CAS Registry Number 190-24-9) was synthesized following a modified route of Rempala et al. [36]. AlCl_3_ (11.4 g, 85.2 mmol) and CuCl_2_ (11.5 g, 85.2 mmol) were suspended in dry CS_2_ (500 mL). Hexaphenylbenzene (1.30 g, 2.43 mmol) was added in two portions over 1 h. The reaction was stirred for 18 h. Methanol (500 mL) was added and the precipitate filtered through a frit. The yellow/brown solid was washed with conc. HCl, water, THF, and CH_2_Cl_2_. The remaining solid was refluxed in toluene (100 mL) for 18 h, filtered, and washed again (toluene, CH_2_Cl_2_) to obtain yellow Hbc (1.27 g, 100%).

MS (LDI): [M]+ calculated *m*/*z* 522.14, found *m*/*z* 521.907.

Cationic and anionic He*_n_*Hbc^±^ ions with *n* ranging up to about 100 were formed as follows: Neutral helium nanodroplets (HNDs) were produced by expanding ultrapure helium (Messer, purity 99.9999%) with a stagnation pressure of 29 bar through a 5 µm pinhole nozzle into ultrahigh vacuum. The nozzle was cooled with a closed-cycle cryocooler (Sumitomo Heavy Industries Ltd., Tokyo, Japan, model RDK-408D2) to 8.5 K (9.65 K for anions). After passing through a skimmer (0.8 mm diameter), the HNDs were ionized by electron impact in a Nier-type ion source, operating at an electron energy of 47.6 eV (19.9 eV for anions) and an electron current of about 0.20 µA.

The resulting beam of highly charged HNDs traversed a pick-up chamber where they captured Hbc molecules evaporated at about 264 °C. Each capture event will release a significant amount of energy, and a large number of helium atoms (about 1600 atoms per eV released) will be evaporated as a result [1]. However, this number pales in comparison to the initial size of the HND, which contains some 10^7^ helium atoms. Subsequently, the doped HNDs collided with an orthogonal metal surface where they splashed and evaporated [14]. The resulting small He*_n_*Hbc^±^ ions (with *n* below about 100) were extracted and guided toward a time-of-flight (TOF) spectrometer system equipped with a reflectron. Additional experimental details have been published elsewhere [37].

### 2.2. Computational Methods

He*_n_*Hbc^±^ cluster ions were modeled using a combination of classical global optimization techniques and PIMD simulations, following earlier works [10,24,34]. Briefly, we used a many-body force field to describe the interaction between the adsorbed atoms or molecules and the hydrocarbon ion, consisting of additive repulsion–dispersion interaction between the helium atoms and the C- or H-atoms of the solvated ion, and a polarizable contribution felt by each helium atom resulting from the distribution of partial charges on the hydrocarbon ion. The parameters for the He-cation individual interactions were determined to reproduce electronic structure calculations at CCSD(T) and MP2 levels on other hydrocarbon molecules, and can be found in [34].

The calculations assume fixed geometries and partial atomic charges on Hbc^±^ ions, whose structures were determined by local optimization from the neutral using DFT at the M06-2X/6-31 + G* level. The resulting geometry and charges are provided in the Appendix A. All quantum chemical calculations were performed using the Gaussian09 software package [38]. As reported previously for neutral Hbc [39], the planar forms were not the local minima for both the cation and anion of Hbc, and the optimized geometries clearly showed ripples that were best seen on the side views of the density plots.

Global optimization by basin-hopping was then performed to identify low-energy structural candidates for 1 to 110 He atoms attached to the hydrocarbon ions, keeping their geometries fixed. For each cluster size, five independent series of 10^5^ local optimizations were carried out and a fictitious temperature of 10 K was employed to evaluate the Metropolis acceptance probabilities. Nuclear quantum effects were subsequently included by performing PIMD simulations and evaluating the virial energy of the system as well as various structural quantities. The PIMD trajectories were carried out at the temperature of 1 K and employed a Trotter discretization number of 128 and a time step of 0.5 fs. These were integrated over 1.2 ns, with the averages being accumulated after 200 ps. Zero-point energy corrections to the static energies of the global minima were also determined in the harmonic approximation, but turned out to be particularly inaccurate to be considered as reliable.

The classical and quantum energies calculated for He*_n_*Hbc^+^ and He*_n_*Hbc^−^ are provided in the Appendix A.

## 3. Results

### 3.1. Experimental Results

A mass spectrum of cationic HNDs doped with Hbc is displayed in Figure 1a. The series of prominent mass peaks, commencing at a mass of 522.141 u and spaced at 4.0023 u, was due to ^4^He*_n_*^12^C_42_^1^H_18_^+^, *n* ≥ 0. Isotopologues of He*_n_*C_42_H_18_^+^ that contain 1, 2, or 3 atoms of ^13^C (natural abundance 1.07%) give rise to weaker satellite peaks; contributions from ions containing ^3^He or ^2^H are negligible. Bare He*_n_*^+^ ions are discernible below 522 u. The mass resolution was about 1500 (measured at full-width-at-half-maximum) in the mass region of Hbc^+^. Two mass peaks caused by impurities are marked by asterisks; they are seen more clearly in the inset in the upper panel and attributed to OHbc^+^ and (H_2_O)_2_Hbc^+^. A more detailed view of the mass spectrum is provided in the Appendix A, where we also compared the current results with previous reports [40,41].

The mass peaks in Figure 1a that are due to He*_n_*Hbc^+^ (containing the main isotopes of H, He, C) are connected by a solid line. The envelope displays notable local anomalies (abrupt drops) at *n* = 14, 38, and 68.

An anionic mass spectrum is displayed in Figure 1b. The series of prominent mass peaks was due to ^4^He*_n_*^12^C_42_^1^H_18_^−^; the weaker series were due to ions containing one or more ^13^C atoms. The mass peaks that were due to He*_n_*Hbc^−^ (containing the main isotopes of H, He, C) were connected by a solid line. The envelope displayed local anomalies at *n* = 14, 26, 60, and 62.

Impurity ions attributed to the H_2_OHbc^−^ and (H_2_O)_2_Hbc^−^ anions are marked in Figure 1b by asterisks (for a detailed view see Appendix A). The arrow at mass 720 u marks a contaminant: In a previous experiment, C_60_ had been vaporized in the oven; residues from that experiment enhanced and broadened the mass peak due to He_49_Hbc^−^ ions that contained two ^13^C atoms. At the same time, the isotopologue of C_60_^−^ that contained two ^13^C atoms (probability 20.7 % relative to ^12^C_60_) superficially enhanced the mass peak assigned to He_50_Hbc^−^ at 722 u.

Local anomalies in the yield of He*_n_*Hbc^±^ ions versus size *n* were identified more reliably when the mass spectrum was processed with the IsotopeFit software, which accounts for isotopic patterns and assigns appropriate contributions of ions that have the same nominal mass [42]. The resulting abundance *I_n_* of cations and anions is plotted versus size *n* in Figure 2a and Figure 3a, respectively. Data for sizes that are affected by the above-mentioned impurities (i.e., *n* = 4 and 9 for cations, *n* = 9 for anions) were omitted. The error bars of the data points were smaller than the symbol size. Additional anomalies were discernible in these distributions, namely at *n* = 26 and 82 for cations, and around 84 for anions.

Anomalies in the ion abundances can be recognized even more clearly if one plots the negative first derivative (−Δ_1_) of the logarithmic ion abundance (i.e., −Δ_1_*I_n_* = ln *I_n_* − ln *I_n_*_+1_); these data are shown in the insets of Figure 2a and Figure 3a for the cations and anions, respectively. As discussed elsewhere [4,43,44], the local maxima in these distributions indicated cluster sizes for which the second derivative of the total energy (i.e., the first derivative of the evaporation energy) was locally enhanced. In other words, for these “magic sizes”, the clusters were particularly stable. The data shown in the insets of Figure 2a and Figure 3a confirm the magic numbers identified thus far; these values are compiled in the first row of Table 1. Values in italics indicate weak anomalies. The data below *n* = 14 are incomplete because of potential contributions from impurities; we disregard possible anomalies in that size range.

### 3.2. Theoretical Results

A comparison of experimental data with atomistic calculations is afforded by the data in Figure 2b and Figure 3b, which show the computed evaporation energies (i.e., the first derivatives of the total energies *E_n_*) for cations and anions, respectively. Full diamonds represent the derivatives of the virial energies; open squares represent the derivatives of the classical energies (without zero-point correction, but the structures themselves may differ from those that minimize the quantum virial energies).

For cations, the quantum evaporation energies exhibited abrupt drops at *n* = 14, 24, 38, and a weaker one at 68. The quantum evaporation energies computed for anions (Figure 3b) also showed stepwise decreases at *n* = 14, 24, 38, 68. Some of these steps (namely at *n* = 68 for cations, and *n* = 24 for anions) are missing in the classical evaporation energies, and another step appeared at *n* = 86. Of course, the classical approach is not likely to correctly describe the properties of He adsorbed on PAHs. All magic numbers observed in the computed evaporation energies are listed in Table 1.

Three of the anomalies (or stepwise decreases) in the calculated evaporation energies of cations (namely those at *n* = 14, 38, and 68) tracked the stepwise decrease in the ion abundance (see Figure 2). The drop at 24 in the calculated data conspicuously missed the drop in the ion abundance at 26. For large sizes, above *n* ≈ 75, the scatter in the computed values was perhaps too large for a meaningful comparison with the experimental data.

For anions (Figure 3), the agreement between the theory and experiment was less convincing; neither the drop at 38 nor that at 68 in the computed evaporation energies were mirrored by anomalies in the ion abundance which instead showed, anomalies at 60 and 62. The computed energies for anions are essentially similar to those obtained for the cations.

Ideally, one would compare the negative first derivative of the logarithmic ion abundance (displayed in the inserts in Figure 2a and Figure 3a) with the second derivative of the computed total energies *E_n_*; these quantities should closely track each other, apart from a scaling factor that depends on the cluster temperature [4,5]. Such a comparison is provided in the Appendix A. However, the scatter in the computed energies (which may suggest incomplete convergence and possibly missed global minima) was amplified when taking the second derivative, and the stepwise drop in the first derivative of *E_n_* was identified more reliably than a corresponding maximum in the second derivative.

Helium atoms bound to graphitic substrates or aromatic molecules tend to be most strongly bound atop the carbon rings, and cluster sizes exhibiting enhanced stability often correlate with an arrangement of the adsorbate layer that is commensurate with the substrate [12,20,22,34,35]. However, a simple model may be misleading because, first, the van der Waals diameter of helium atoms (about 0.3 nm) significantly exceeds the spacing between adjacent hollow sites of the honeycomb lattice (0.246 nm). Second, a weakly corrugated surface may fail to localize the helium atoms, even at 0 K [18].

It is illuminating to inspect the computed structures of *n* He atoms adsorbed on Hbc^±^. For the quantum calculations, the energetically favored arrangement of helium on even-numbered complexes is generally of the form ½ (*n*, *n*) (i.e., exactly one-half of all atoms resides on either side). The 10-mer, with the arrangement (6, 4) for either charge state, is the only exception. For the energetically favored arrangement of helium on odd-numbered clusters, one side of the molecule adsorbs one atom more than the other side.

Figure 4 and Figure 5 display the computed quantum structures for selected sizes, 2 ≤ *n* ≤ 74; structures for all cations of size *n* ≤ 90 are provided in the Appendix A. For each size, the helium densities obtained from the PIMD simulations were superimposed on the structure of Hbc^±^. Cations and anions are depicted in the top and bottom rows, respectively. By and large, the cations and anions showed very similar structures, even though some residual differences will be pointed out. For the 38-mer in Figure 4, we also show the side views; side views of larger cationic clusters are included in Figure 5. Side views of the anions were not included because they were very similar to those of the cations.

For *n* = 2, the He atoms were localized on either side of the central ring. For *n* = 6, they were delocalized over the six rings adjacent to the unoccupied central benzene ring. This ring remains unoccupied until it fills again, starting at the 10-mer.

The magic anionic 14-mer shows the strong localization of one atom each on either side of the central benzene ring plus six atoms each above and below the six outer benzene rings, preserving the approximate sixfold symmetry of the substrate. The arrangement was similar to that of the commensurate √3 × √3 phase on graphite in which He atoms are adsorbed in the hollows of the second-nearest carbon rings [45]. The cationic He_14_Hbc^+^ also formed a pronounced magic number in the mass spectra, but the computed degree of localization was weaker.

The cationic 26-mer, which was (weakly) magic in the experimental data, had all atoms strongly localized, the central benzene ring was occupied, and the pattern exhibited sixfold symmetry (the degree of localization in the anion is less pronounced). The 14 innermost atoms occupied positions similar to those in the 14-mer; the additional 6 + 6 atoms in the second helium ring were localized on the incomplete carbon rings saturated with peripheral hydrogen.

In the calculations, the 24-mer rather than the 26-mer appeared to be magic for either charge state. The degree of localization was notably weaker than in the 26-mer (especially the cationic 26-mer), and the arrangement exhibited 3-fold rather than 6-fold rotational symmetry. Perhaps the most striking difference between the structures computed for the 24- and 26-mers was the vacancy at the central ring of the 24-mers.

Upon further growth of the adsorbate layer, the He atoms became more delocalized, but the pattern sharpened again at *n* = 38, which was the next magic number in the mass spectra of cations, although it did not for the anions. The computed structure of the 38-mer resembled that of the 26-mer, with an additional six atoms forming a third, slightly larger ring. Close comparison with the 26-mer revealed another difference: The 6 + 6 atoms in the first helium ring moved inward, and they occupied bridge positions between the outer benzene rings rather than the hollows of the latter.

Thus far, the observed magic numbers 14, 26, and 38 have been found to follow a simple pattern: After decorating the central ring with 1 + 1 helium atoms, 6 + 6 atoms are added successively to form larger and larger helium rings; the arrangement of the atoms in the rings can conceivably preserve the symmetry of the substrate. Adding further rings of 6 + 6 atoms, one will anticipate magic numbers at *n* = 50, 62, 74, etc.; these numbers are listed in the last row of Table 1. The 50-mer did not form a magic number in the mass spectra, but the 62-mer did in the anionic spectrum.

Models based on sphere packing may, of course, produce a fortuitous agreement. The filling of a fourth ring of helium atoms started above *n* = 38, as indicated by the halo in Figure 5 that surrounds the cationic 40-mer. This ring remained circular and diffuse (indicating delocalization within the ring) up to about *n* = 62. The computed arrangement of the 38 helium atoms in the first three rings of the 62-mer was markedly similar to that in the 38-mer; the additional 12 + 12 atoms formed one additional rather than two distinct helium rings.

The 68-mer formed a weak magic number in the PIMD data for both charge states, in agreement with the cationic but not the anionic mass spectra. Side views of the density plots, given in Figure 5 for the cations, showed that at this size, the additional helium atoms were close to the average symmetry plane of Hbc^±^, rather than being distributed in a 3 + 3 arrangement among the two helium layers. As more helium atoms were added, they still lay near the symmetry plane but were pushed away from the center and produced an hexagonal pattern at *n* ≈ 74, above which the arrangement became more irregular. The side views of the density plots revealed that this was caused by the occasional popping of helium atoms away from the first solvation layer, found, for example, at size 76 (see the Appendix A). Above *n* = 85, the helium atoms were consistently found in a second layer above and below the first solvation layer. The extrusion of atoms from the first layer to form floaters also occasionally took place in the range *n* = 70–80, which explains the more significant fluctuations in the virial energies perceivable in Figure 2 and Figure 3.

In Figure 6, we compare the structures computed for He*_n_*Hbc^+^, *n* = 14, 26, 38, and 68 using the PIMD and the classical approach (top and bottom rows, respectively). One general difference between the classical and quantum structures arises from the size mismatch between helium atoms and the spacing between adjacent hexagons. The energetic cost of a 1 × 1 arrangement in which adjacent benzene rings were covered was less severe in the classical calculation. This was best seen for *n* = 14, where the adsorbates were nearly arranged in a 1 × 1 motif, while they preferred a √3 × √3 pattern once vibrational delocalization was accounted for in the PIMD approach. However, in both cases, the 14-mer had approximate *D*_6*h*_ symmetry and enhanced stability.

For the 38-mer, the results of the classical calculation showed no surprises: it had enhanced stability, and the arrangement of atoms was similar to the 38-mer computed with the PIMD approach, except for the reduced distance of atoms in the first helium ring from the center.

With the PIMD approach, all even-numbered clusters, except for the 10-mer, preferred a (½ *n*, ½ *n*) arrangement of the helium layer. This is less generally true with the classical approach, especially near the sizes of 10 to 15, in which it is highly favorable to have seven atoms on one side forming a filled hexagon, leading to stable classical structures of the (7, *n* − 7) form in this range. While the 26-mer was still of the (13, 13) type, this was not the case for the 68-mer, which was found as (35, 33) in the classical approximation, and was restored as (34, 34) once the zero-point effects were included. The higher symmetry discerned for many helium densities in Figure 4, Figure 5 and Figure 6 is thus the result of the quantum nuclear effects.

It is also useful to compare the present results for He*_n_*Hbc^+^ with those obtained previously for He adsorbed on coronene. The experimental magic numbers for He*_n_*Cor^+^ were *n* = 38, 41, and 44; neither 14 nor 26 were magic [20]. The absence of a magic number for the 14-mer may seem surprising, given its prominence in our mass spectra of He*_n_*Hbc^+^, and in a classical study of neutral He_14_Cor, which showed that the He atoms were adsorbed in the hollows of the seven benzene rings [21]. However, according to our quantum calculations (see Figure 4), the He atoms in the first ring of He_14_Hbc^+^ resided far away from the molecular C_6_ axis, in a ring that passed through the dimples of the outer benzene rings rather than the inner benzene rings. Thus, Cor^+^ simply does not offer adsorption sites for He atoms in the putative first ring, which resemble those in Hbc^+^.

## 4. Discussion and Conclusions

We investigated the adsorption of He on the charged Hbc. This nearly planar molecular substrate was significantly larger than other planar PAHs that have been studied previously. It is the first PAH for which magic numbers in mass spectra indicate the successive closure of several subshells. Furthermore, the present work compared, for the first time, the adsorption of He on the cationic and anionic complexes. Although the PIMD calculations did not correctly reproduce every magic number observed in the mass spectra, they do illuminate the peculiar growth of the He adsorption layer on a large PAH, which here was largely driven by its (approximate) sixfold symmetry. After filling the first two preferred adsorption sites above and below the central benzene ring, the first three ring-shaped subshells contain 6 He atoms each above and below the molecular plane, which endow complexes containing 14, 26, or 38 with enhanced stability. The He atoms in filled rings are localized, or at least localized within each ring, while atoms in partially complete subshells are rather delocalized. The radii of the first three rings shrink as successive rings are filled, but beyond *n* = 38, the arrangement of He in the first three rings remains frozen.

The mass spectra indicate some marked differences between the two charge states: closure of a subshell at *n* = 38 was observed for cations but not anions, and another subshell closed at *n* = 68 for the cations but at 60 and/or 62 for the anions. The PIMD calculations suggest that the first solvation shell closes at or around *n* = 68, in agreement with the measurements. However, they failed to explain the higher stability of the 26-mer seen in the mass spectrum, although the quantum structure found at this size was particularly symmetric.

The main discrepancy between the experiment and theory thus amounts to the anionic systems, but is such that the validity of the modeling for anionic systems could be questioned altogether. By construction, the polarizable potential used in the simulations assumes that the electron density of the molecular ion felt by helium atoms is correctly described by partial charges placed on the hydrocarbon atoms. While this approximation is reasonable for cations, the excess electron in anions is expected to have a more delocalized character that the atomic charges model may fail to capture. In this respect, it would be useful to consider a refined version of the atomistic description of Hbc^−^ with additional sites to better represent the electrostatic potential in the vicinity of the approximate molecular plane. Such a model should be parametrized using electronic structure calculations that employ a larger basis set than the rather simple one (6-31 + G*) used here, and especially with diffuse functions. The reference ab initio calculations between helium and the hydrocarbon ions could also be revised in the light of the possible influence of the relativistic and quantum electrodynamics effects [46]. It is also possible that the current version of the polarizable potential, which is restricted to interactions between induced dipole moments but without self-consistent dipoles, is too crude for anions.

Another possible source of error in the modeling is the rigid assumption for the Hbc^±^ ions. At the presently used level of theory, the minimized structures have vibrational modes in the 10–20 cm^−1^ range that could be sensitive to the presence of helium atoms, and could even be delocalized themselves to some extent. The PIMD simulations with flexible hydrocarbon cations would not be feasible at the DFT level, but any simpler model would still have to reproduce the non-planar equilibrium geometries of the bare ions.

To assess models for anions, additional measurements for helium adsorbed on anionic hydrocarbons or fullerenes could be particularly insightful. Experiments with negatively charged acenes are in progress.

## Figures and Tables

**Figure 1 molecules-27-06764-f001:**
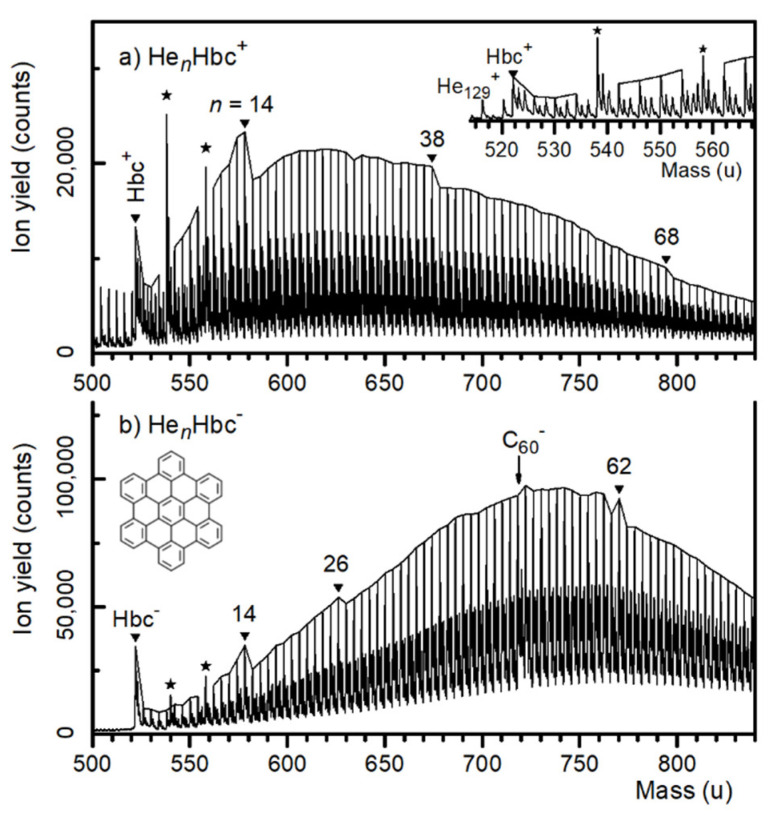
(**a**) Mass spectrum of hexabenzocoronene (Hbc) cations complexed with helium. The inset displays a narrow section of this spectrum. Mass peaks arising from contaminants are marked by asterisks. Mass peaks due to the main isotopologue of He*_n_*Hbc^+^ are connected by a solid line; local anomalies are indicated. (**b**) Similar to panel (**a**) for negatively charged ions.

**Figure 2 molecules-27-06764-f002:**
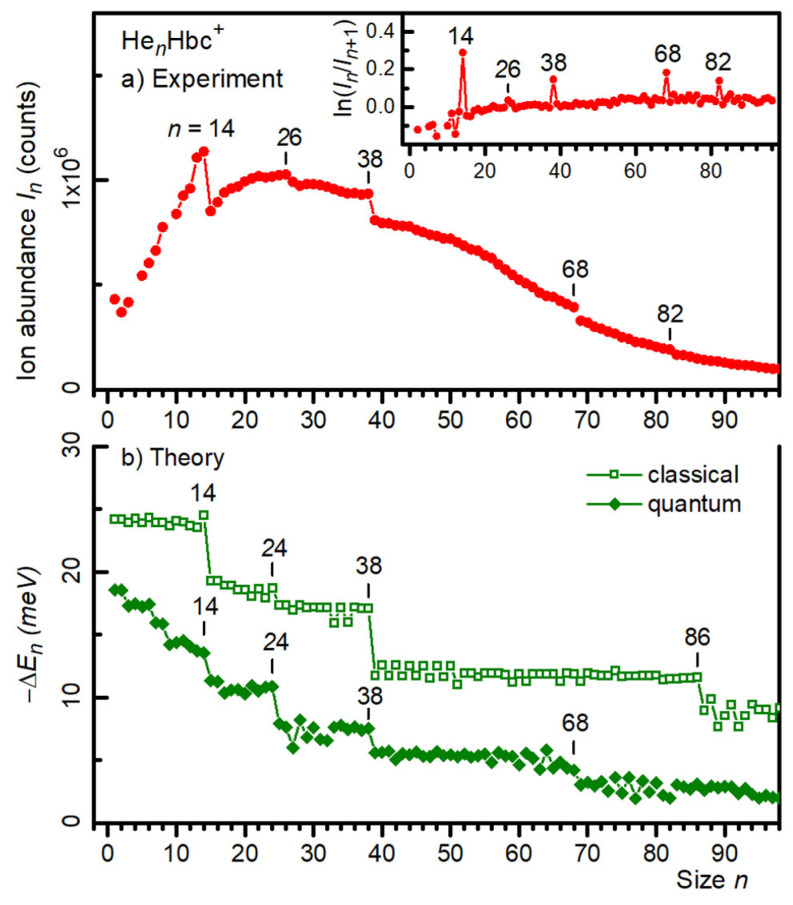
(**a**) Ion abundance *I_n_* of He*_n_*Hbc^+^ cations versus size *n*. The insert displays the negative first derivative of the logarithmic ion abundance. (**b**) The first derivative of the energy −*E_n_* of He*_n_*Hbc^+^ calculated from the classical global minima and the quantum virial energy (open squares and filled diamonds, respectively).

**Figure 3 molecules-27-06764-f003:**
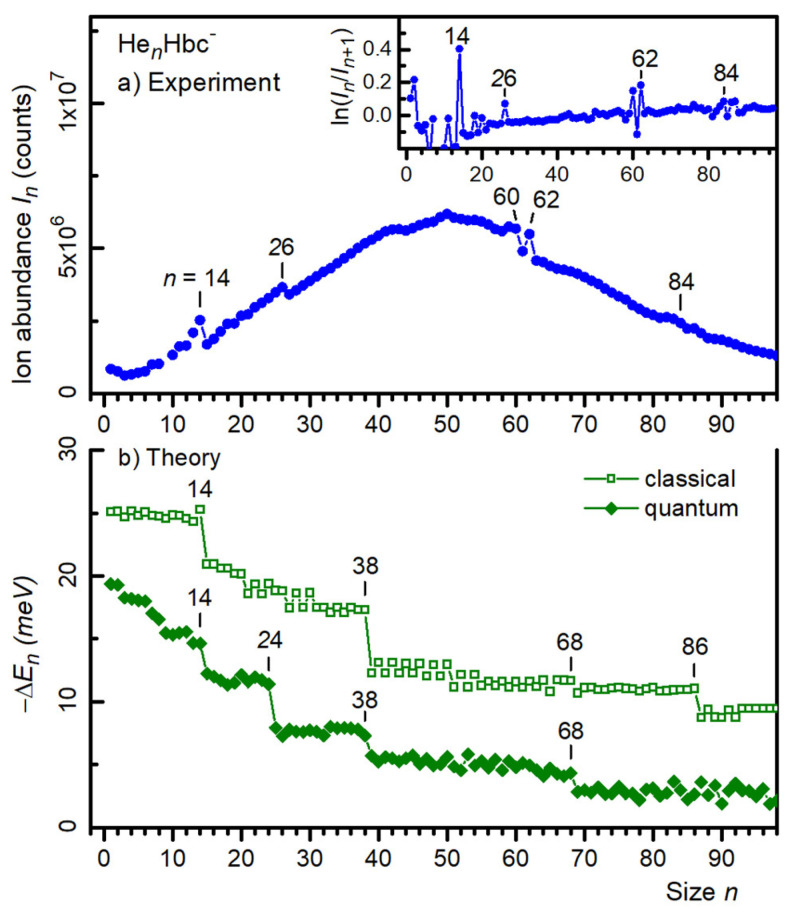
(**a**) Ion abundance *I_n_* of He*_n_*Hbc^−^ anions versus size *n*. The insert displays the negative first derivative of the logarithmic ion abundance. (**b**) The first derivative of the energy −*E_n_* of He*_n_*Hbc^−^ calculated from the classical global minima and the quantum virial energy (open squares and filled diamonds, respectively).

**Figure 4 molecules-27-06764-f004:**
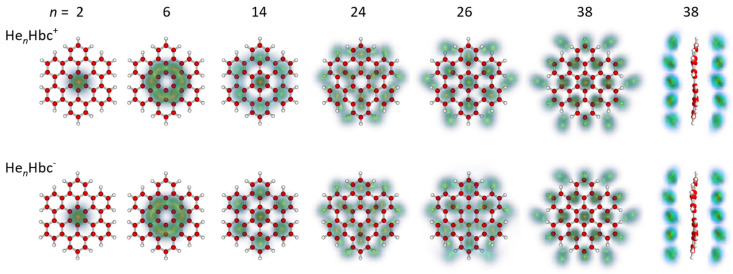
Selected structures of the He*_n_*Hbc^+^ and He*_n_*Hbc^−^ cluster ions (top and bottom row, respectively); the size *n* is indicated on top. For each size, the helium densities obtained from the PIMD simulations are superimposed on the structure of Hbc^±^. Side views of the 38-mers are shown on the right.

**Figure 5 molecules-27-06764-f005:**
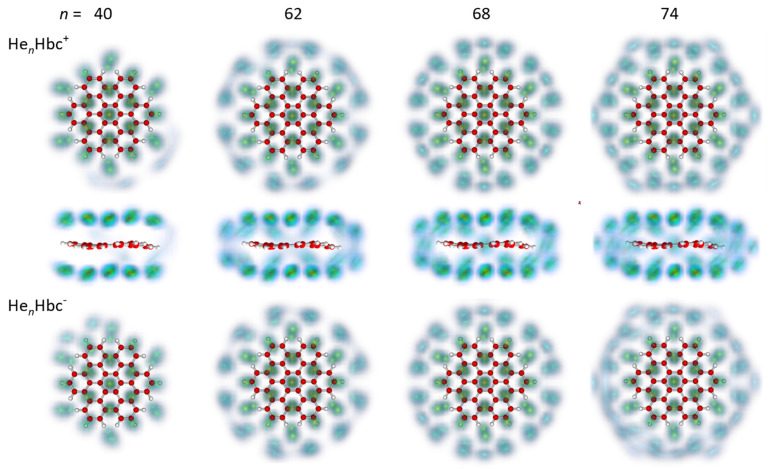
Selected structures of the He*_n_*Hbc^+^ and He*_n_*Hbc^−^ cluster ions (top and bottom row, respectively); the size *n* is indicated on top. For each size, the helium densities obtained from the PIMD simulations were superimposed on the structure of Hbc^±^. Side views of the cationic clusters are shown beneath the corresponding front views.

**Figure 6 molecules-27-06764-f006:**
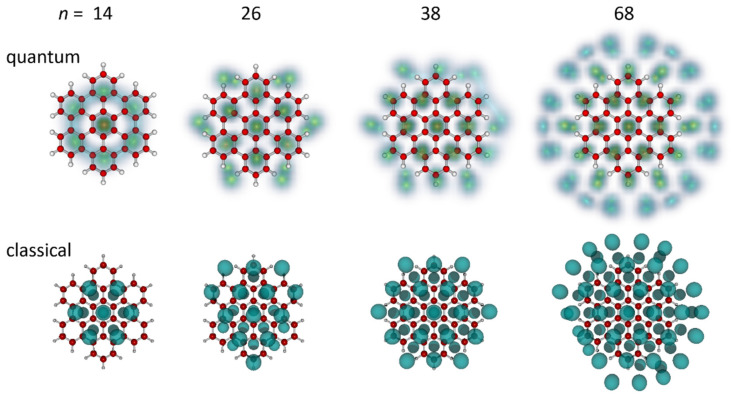
Structures of He*_n_*Hbc^+^ for *n* = 14, 26, 38, and 68 obtained in the quantum and classical calculations (top and bottom row, respectively).

**Table 1 molecules-27-06764-t001:** The magic numbers observed and calculated for He*_n_*Hbc^±^, *n* ≥ 14. Minor magic numbers are in italics. The last row lists the shell closures that are expected if the adsorbate layer grows by successively filling rings containing 6 + 6 He atoms each, and the central benzene ring is occupied. Underlined numbers indicate an agreement between the experiment and quantum calculations.

	Cations	Anions
Experiment	14, 26, 38, 68, 82	14, 26, 60, 62, *84*
Theory (PIMD)	14, 24, 38, *68*	14, 24, 38, 68
Theory (classical)	14, *24*, 38, 86	14, 38, *68*, 86
Sphere packing	14, 26, 38, 50, 62, 74	14, 26, 38, 50, 62, 74

## Data Availability

The data presented in this study are available in the Appendix A.

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
