# Peer review of "Solvation of Large Polycyclic Aromatic Hydrocarbons in Helium: Cationic and Anionic Hexabenzocoronene"

_molecules, 2022, doi:10.3390/molecules27196764_

Round 1

Reviewer 1 Report

In this study, the authors examined the adsorption of helium on charged hexabenzocoronene. The manuscript is clearly written, the methods and results are concise and well described. My recommendation is to correct the minor flaws that have been noticed in the manuscript.

1. Was the mass of 11.4 g of AlCl3 and 11.5 g of CuCl2 added in a liter of solution or 500 ml to obtain the appropriate molarities?

2. Add information about the manufacturer and country of manufacture to the used devices.

3. After the abbreviation for the compound has been introduced into the text for the first time, use the abbreviation further on in the text.

4. Please supplement the reference list with more recent references.

Reviewer 2 Report

The manuscript "Adsorption of Helium on large, planar polycyclic aromatic hydrocarbons: Cationic and anionic exabenzocoronene" describes the results of experimental and theoretical studies on the interaction between model planar polycyclic aromatic hydrocarbon and helium nanodroplets in low temperatures. The manuscript contains significant results, and I am fully convinced that it has to be published. I like figures 4 and 5. However, the current version of the manuscript requires improvements. I have one significant objection. The Authors used M06-2X/6-31+G*. The basis set is relatively small, and the M06-2X has not been corrected by empirical dispersion. The empirical dispersion correction is essential for the week interaction, like He-He or PAH-He. For these reasons, I have doubts about the quality of theoretical results. The best way is the application of high quality ab initio method with a huge basis set; however, this is not feasible for such an extensive system. I recommend doing some test computations for a slightly smaller system and putting them in the context of previous benchmark studies on reproducing energies of the nonbonding interactions by different electronic structure methods (there are several of them, the selected examples are the following: J. Chem. Theory Comput.2005, 1, 3, 415–432, J. Chem. Theory Comput.2011, 7, 1, 88–96, Chem. Rev.2016, 116, 9, 5038–5071). How does the expected error of interaction energies translate into the uncertainty of obtained results? Do the possible weaknesses of the calculation method explain some inconsistencies between the experiment and the theory? How much do other approximations affect the results? I am curious how the Authors will comment on the impact of relativistic and QED effects on the obtained results. For collisions at very low-temperature, they may be of importance (Phys. Rev. A 102, 020801(R)). Could it be similar here?

I also have some less important suggestions. Overall, the text is of acceptable quality, but I encourage authors to try to work on the style of the manuscript, especially the Introduction. Additionally, please

1) add some comments about the references (or identify a better place for them) in part: "A more detailed view of the mass spectrum is provided as Supplementary Material (Figure S1a) [38-39]."

2) improve the formation of superscripts. Currently, they are too low in comparison to the remaining text. 

3) put coma instead of the dot after 14 on page 7 in fragment "n = 14. 24, 38, "
